

# Skills in sub-seasonal to seasonal terrestrial water storage forecasting: insights from the FEWS NET land data assimilation system

Bailing Li[1], Abheera Hazra[1], Amy McNally[2], Kimberly Slinski[1], Shraddhanand Shukla[3], Weston Anderson[4]

[1]ESSIC University of Maryland, College Park, MD 20740, USA

[2]NASA Goddard Space Flight Center, Greenbelt, MD 20771, USA

[3]University of California at Santa Barbara, Santa Barbara, CA 93106, USA

[4]Department of Geography, University of Maryland, College Park, MD 20740, USA

*Correspondence to: Bailing Li (bailing.li@nasa.gov)*

**Abstract.** Accurate prediction of terrestrial water storage (TWS), the sum of soil moisture, groundwater, snow/ice, and surface water, is critical for informing water resource management and disaster responses. In this study, we evaluated subseasonal to seasonal (S2S) TWS forecasts, produced by the FEWS NET land data assimilation system (FLDAS), over Africa using observations from the Gravity Recover and Climate Experiment (GRACE) and its Follow-On (GRACE/FO) mission. FLDAS consists of two advanced land surface models, Noah-MP and the NASA Catchment Land Surface Model (CLSM), both of which simulate key TWS components including groundwater. Results show that CLSM is more skillful in forecasting TWS anomalies at S2S scales than Noah-MP, with >0.6 relative operating characteristics (ROC) scores over more than half of the study domain across the 1-6 months lead times. CLSM forecasts also maintain stronger correlations with GRACE/FO data than Noah-MP, particularly at longer lead times, owing to more skillful reanalysis-based initial conditions and stronger persistence in simulated TWS. In contrast, Noah-MP forecasts show weaker skill, especially in central Africa where the skill also declines rapidly with lead time.

Evaluation results show that accuracy of TWS forecasts is strongly influenced by precipitation interannual variability: forecasts driven by precipitation products with lower precipitation interannual variability are generally more accurate than those driven by higher precipitation variability. The performance gap between Noah-MP and CLSM is also more pronounced in regions with higher precipitation variability such as central Africa. This sensitivity arises because TWS often exhibits strong multi-year variability in responses to interannual precipitation, making realistic simulation of long-term variability critical for skillful TWS forecasts. The superior performance of CLSM is attributed to its strong representation of upward groundwater movement, especially during prolonged droughts, which enhances TWS interannual variability. In contrast, the weak representation of capillary rise in Noah-MP limits its ability to capture effects of long-term precipitation variability on TWS. Both models





exhibit lower correlation and higher RMSEs when evaluated against GRACE/FO data than relative to reanalysis, further underscoring substantial uncertainty in model physics.

Autocorrelation analyses show that TWS persistence is closely linked to groundwater persistence. CLSM groundwater exhibits stronger persistence than that of Noah-MP, owing to its ability to simulate groundwater responses to long-term precipitation variability. While persistence provides an important source of predictability, our results also show that inaccurate persistence, such as that associated with anthropogenically induced trends and changes in precipitation that are often inadequately captured by land surface models, can degrade forecast skill. These findings underscore the importance of using independent datasets
such as GRACE/FO observations to evaluate TWS forecasts.

## 1 Introduction

Changes in terrestrial water storage (TWS), the sum of soil moisture, groundwater, snow/ice and surface water, reflects cumulated impacts of precipitation and evapotranspiration over weeks to months (Humphrey et al., 2016). As such, it provides unique insight into hydrological extremes (floods and droughts) and their responses to climate variability and climate change
(Zhao et al., 2017; Rodell & Li, 2023; Li & Rodell, 2023; Li et al., 2025). Skillful TWS forecasts at subseasonal to seasonal (S2S) scales are therefore of great value for providing early warnings on water shortage and crop failure, especially in Africa, where persistent food and water insecurity faced by many communities are often exacerbated by frequent floods and droughts (Scanlon et al., 2020; Ngcamu & Chari, 2020; Cook et al., 2021; WMO, 2025).

Thus far, most studies have focused on evaluating TWS forecasting skills by climate models at decadal scales (e.g., Jensen et al., 2020; Yuan and Zhu, 2018; Zhu et al., 2019). These evaluations typically compare initialized forecasts, where initial conditions are derived from model simulation driven by observation or reanalysis-based atmospheric forcing data, with uninitialized ones to obtain skill scores. Initial conditions have been found to provide more skill than dynamical climate forecasts alone in 1-4 years lead time, suggesting persistence of TWS as a key source of predictability (Zhu et al, 2019).
However, since most climate models do not simulate groundwater, the reported persistence mainly reflects that of soil moisture. More importantly, in the absence of independent observational datasets, such evaluations may overestimate the role of persistence as they fail to account for uncertainties in land surface model physics and meteorological forecasts.

Groundwater, located in the deeper subsurface, has longer memory than other near surface processes such as soil moisture,
making it a potential source of predictability for TWS forecasting (Eltahir and Yeh, 1999; Li et al., 2015). However, modeling groundwater is subject to greater uncertainty due to lack of information on hydrogeological properties and observational data to constrain simulation of deep subsurface processes (Xia et al., 2017). As a result, reanalysis-based groundwater estimates, when used as initial conditions, may not deliver correct persistence or memory for enhancing forecast skill. Furthermore,



because of its long memory, groundwater is more sensitive to biases in the meteorological forecasts that drive TWS forecasts.

Previous studies have shown biases in S2S precipitation forecasts vary depending on climate conditions and terrains (Shukla et al., 2019; Slater et al., 2019; Zhang et al., 2021; Levey and Sankarasubramanian, 2024; Phakula et la., 2024). However, examining groundwater responses to meteorological forecasts is hindered by the scarcity of in situ groundwater observations at the continental to global scales (Jasechko et al., 2024).

TWS observations from the Gravity Recovery and Climate Experiment (GRACE) and its Follow On (hereafter GRACE/FO, Landerer et al., 2020) mission provide a unique opportunity to evaluate S2S TWS forecasts. Representing vertically integrated water storage changes, GRACE/FO data exhibits strong temporal variabilities from subseasonal to interannual scales depending on climate conditions (Humphrey et al., 2016). While sub-seasonal variability is essential for assessing S2S forecast skill, interannual variability is equally important for establishing robust climatology needed for forecasting TWS anomalies.

GRACE/FO data have been widely used to validate reanalysis estimates and to identify deficiencies in model physics in large-scale hydrological models (e.g., Döll et al., 2014; Scanlon et al., 2018; Bonsor et al., 2018; Li et al., 2019a). However, few studies have used GRACE/FO data to evaluate TWS forecasts. Cook et al. (2021) assessed TWS forecast skill over Africa using a reconstructed GRACE product. With more than two decades of nearly continuous observations, GRACE/FO observations are ideal for objectively assessing S2S TWS forecast skill and exploring factors influencing TWS predictability.


The hydrological forecasting system, FEWS NET Land Data Assimilation System Forecast (FLDAS-Forecast), was developed to provide early warnings on droughts and floods across Africa (Arsenault et al., 2020; Hazra et al., 2023). FLDAS-Forecast is a custom instance of the NASA Land Information System (LIS), an advanced computing framework that supports land surface modeling and data assimilation (Kumar et al., 2006). FLDAS-Forecast comprises two advanced land surface models,

Noah-MP and the NASA Catchment Land Surface Model (CLSM), both of which simulate major TWS components including groundwater. FLDAS-Forecast ingests precipitation forecasts from the full North American Multi-Model Ensemble (NMME, Kirtman et al., 2014) which has shown to improve soil moisture forecasts in southern Africa, compared to forecasts based on a single NMME model (Hazra et al., 2023).

The primary goal of this study is to provide an objective evaluation of the skill of S2S TWS hindcasts from FLDAS-Forecast, using GRACE/FO observations. By leveraging the multi-model framework of FLDAS-Forecast and a full ensemble of NMME meteorological forecasts, the evaluation aims to improve understanding of how model physics employed by land surface models influence TWS forecast skill and how they interact with meteorological forecasts. To isolate the impact of model physics from those of initial conditions and meteorological forecasts, TWS hindcasts were also evaluated using reanalysis

based TWS estimates, which are used as initial conditions for TWS forecasts. Unlike past studies where S2S forecasts were evaluated by seasons, statistics analyses are performed over the entire study period (2003-2020) to better examine the role of long-term variability and persistence of TWS processes in TWS forecasts. Autocorrelation analysis is performed for different





TWS processes to examine the role of persistence on forecast skill and to assess the relative contribution of each process to overall TWS persistence.

## 2 Data and evaluation metrics

### 2.1 Observational and reanalysis-based meteorological input

Precipitation from the Climate Hazards Infrared Precipitation with Stations (CHIRPS, Funk et al., 2015) and other meteorological fields from the Modern-Era Retrospective Analysis for Research and Applications, Version 2 (MERRA-2, Gelaro et al., 2017) are used to drive model simulation by Noah-MP and CLSM from 1982 to present. The output of these simulation runs is then used as initial conditions for issuing TWS hindcasts (Hazra et al., 2023).

CHIRPS integrates satellite-based precipitation estimates with station data to produce global precipitation time series at 0.05° spatial resolution and a 6-hour interval. MERRA-2, which has 0.5° in latitude by 0.625° in longitude resolution, is an atmospheric reanalysis product based on the Goddard Earth Observing System (GEOS) model, featuring assimilation of various atmospheric observations such as radiances, surface winds, temperature, and aerosol and improved representation of stratosphere and cryosphere processes (Gelaro et al., 2017).

### 2.2 Meteorological hindcasts

FLDAS-Forecast uses precipitation hindcasts from a suite of NMME models (Table 1) and non-precipitation fields from GEOS (Borovikov et al., 2019) to generate S2S TWS hindcasts. Since GEOS has fewer ensemble members than NMME models, GEOS ensemble members are randomly selected to pair with the NMME models for generating TWS forecasts (see details in Hazra et al., 2023).

The NMME hindcasts are provided as monthly data on a 1° resolution global grid and are bias-corrected and spatially downscaled to the 0.25° resolution using algorithms implemented in FLDAS-Forecast using higher-resolution precipitation data from CHIRPS (Arsenault et al., 2020; Hazra et al., 2023). Monthly hindcasts are first downscaled to daily values using daily GEOS data and further downscaled to 6-hour intervals using MERRA-2 sub-daily climatology (Arsenault et al., 2020). For simplicity, the combined NMME and GEOS meteorological hindcasts are referred to as NMME models in the following sections.



Table 1. NMME model specifics used in FLDAS-Forecast

| Models | Centers | ensemble members |
|---|---|---|
| CFSv2 | NOAA/NCEP | 12 |
| CESM1 | NCAR | 10 |
| CanESM5(CSM5) | Environmental Canada | 10 |
| GEOSv2 | NASA/GMAO | 4 |
| GFDL | NOAA/GFDL | 15 |
| GEM5.2-NEMO(GNEMO5.2) | Environmental Canada | 10 |

## 2.3 Land surface models and TWS hindcasts

Both Noah-MP and CLSM simulate key components of TWS, soil moisture, groundwater storage and snow water equivalent (SWE), based on water and energy balance equations.  However, they differ considerably in model physics, particularly in subsurface water flows (see Table 1 of Xia et al., 2017 for model configuration and descriptions).

Noah-MP simulates soil moisture dynamics in four unsaturated soil layers based on Richards' equation (Niu et al., 2011).  Groundwater storage in FLDAS-Forecast/Noah-MP is represented by a linear reservoir scheme that computes groundwater
storage changes based on net water exchanges between the lowest soil layer and the aquifer (Niu et al., 2011).  Although the scheme simulates capillary rise, the upward water movement from the aquifer to the upper unsaturated soil, is minimal, resulting in small seasonal variations in simulated groundwater in some regions (Xia et al., 2017; Li et al., 2021).

In contrast, CLMS simulates water storage changes at three water storage layers: a 2 cm surface layer, a 1 m root zone and the total profile (Koster et al., 2000).  The depth of the soil profile is determined by a spatially varying bedrock depth parameter
(see Fig.10 of Li et al., 2019b for the spatial map).  Water flows among these layers are governed by empirically derived time constants that actively redistribute water, transferring water downward during precipitation events, and upward during the dry months to sustain ET.  This strong coupling between surface and deep layers results in pronounced seasonal variations in CLSM simulated groundwater and TWS, even in dry climates (Xia et al., 2017; Li et al., 2019b).  While it does not explicitly model groundwater, CLSM groundwater storage can be obtained by subtracting water storage in the root zone from that of the
soil profile.

The two models also employ different physics for ET estimates which, along with precipitation, are major controls on temporal variability of groundwater (Eltahir and Yeh, 1999; Li et al., 2015).  Previous studies have shown that CLSM tends to simulate higher ET than other land surface models, primarily due to its strong coupling among soil layers and the specific ET algorithms it employs (Xu et al., 2019).  For instance, bare soil evaporation is computed as a nonlinear function of soil moisture in Noah-
MP, but a linear function in CLSM (Niu et al., 2011; Koster and Suarez, 1996).  Although both models employ the TOPMODEL




concept to simulate surface and subsurface (baseflow) runoff, discrepancies in simulating ET and profile moisture lead to different runoff estimates by the two models (Xia et al., 2017).

Neither model simulates surface water, which is detected by GRACE/FO satellites. However, surface water contribution to TWS is generally smaller compared to other TWS components, except in areas with large surface water bodies such as African Great Lakes, the Kariba reservoir and other reservoirs along the Nile (Rodell et al., 2002; Getirana et al., 2017). Additionally, because snow is negligible in Africa, simulated TWS in this study is thus represented as the sum of soil moisture in the unsaturated zone, 2 m for Noah-MP and 1m for CLSM, and groundwater storage.

TWS hindcasts with lead times of 1 to 6 months were generated by forcing Noah-MP and CLSM with the NMME hindcasts described above. As discussed above, initial conditions for each hindcast at any month are obtained from the corresponding model simulation driven by CHIRPS precipitation and non-precipitation fields from MERRA-2. Since MERRA-2 is a reanalysis product, these simulations are referred to as reanalysis in the following sections.

## 2.4 GRACE/FO TWS observations

GRACE/FO data used in this study were developed by the Center for Space Research (CSR) at the University of Texas based on the mass concentration (mascon) approach (Save et al., 2016). The mascon approach utilizes time-variable constraints to constrain the inversion of satellite ranging data to gravity fields at each mascon block. The mascon approach eliminates the need for postprocessing as with the spherical harmonical approach and thus better preserves signals related to TWS changes (Landerer & Swenson, 2012; Save et al., 2016).

CSR GRACE TWS observations are provided as monthly anomalies relative to the 2004-2009 temporal mean, at a 0.25° spatial resolution. However, the effective resolution remains relatively coarse, approximately 150,000 km$^2$ at mid-latitudes (Tapley et al., 2004). There are 34 months with missing data, including the 11-month gap between the two missions. Missing data were filled using linear interpolation, except for the 11-month gap. We found filling the gap had no noticeable impact on the statistical results.

## 2.5 Data processing and study domain

To ensure consistency with GRACE/FO data, the temporal mean of simulated TWS over 2004-2009 was removed at each grid cell. Next, the monthly mean, one for each calendar month, was removed to obtain TWS anomalies for both simulated TWS and GRACE/FO data. For evaluation, TWS time series for the overlapping period between TWS hindcasts and GRACE/FO data, 2003-2020, were extracted from both the reanalysis and the hindcasts at 2-,4- and 6-month lead times at each grid cell.

The FLDAS-Forecast domain encompasses the African continent and a large portion of the Middle East (Supplementary Fig.S1). Northern Africa and parts of the Middle East have experienced long-term TWS declines associated with extensive groundwater withdrawals for irrigation (Gossel et al. 2004; Rodell et al., 2018; Frappart et al.,2020; Scanlon et al., 2020). Since FLDAS-Forecast does not simulate these anthropogenic effects, these regions were excluded from the evaluation using the groundwater depletion masks provided by Rodell et al. (2018).

For drought and flood monitoring, percentiles are obtained by ranking forecasts against climatology derived from hindcasts for 2003-2020 for each NMME model. Average percentiles across all NMME models are then used to produce percentile maps. In addition, probabilities are computed for tercile forecasts, below normal (< 33%), normal (33% - 67%) and above normal (>67%), across all ensemble members at each grid cell (see details in Hazra et al, 2023).

## 2.6 Evaluation metrics

The root mean square error (RMSE) and Pearson correlation are used to evaluate TWS anomalies, with respect to GRACE/FO TWS anomalies. Additionally, the relative operating characteristic (ROC) score, representing the ratio of hit rates to false alarm rates, is used to assess tercile-based TWS hindcasts (Met Office). A ROC score of 1 indicates a perfect forecast, while scores below 0.5 suggest no skill (Met Office). High ROC scores and strong correlation are commonly interpreted as indication of skillful forecasts (e.g., Yuan and Zhu, 2018).

## 3 Results

The skill of TWS forecasts is influenced by three factors, initial conditions, meteorological forecasts and physics employed by the land surface model. To isolate contribution of each factor, we first examine temporal variability of reanalysis of TWS processes. We then evaluate hindcasts at different lead times using GRACE/FO data to explore different controls on TWS accuracy. To isolate the impact of model physics, we further compare TWS hindcasts with the reanalysis. Finally, we analyze TWS persistence and explore its role in influencing TWS forecast skill.

### 3.1 Evaluation of reanalysis

Domain average estimates of reanalysis show large discrepancies between the two models at both seasonal and non-seasonal timescales (Fig.1). Noah-MP, with a 2 m soil depth, simulates greater soil moisture variability than CLSM which has a 1 m soil depth. In contrast, CLSM simulates much stronger groundwater variations than Noah-MP across both seasonal and non-seasonal scales (Figs.1c,d), with the seasonal amplitude and temporal standard deviation of non-seasonal TWS being nearly five times larger than those of Noah-MP (Figs.1e,f).

As a result of the strong groundwater temporal variability, non-seasonal CLSM TWS estimates also show strong temporal variation that contributed to stronger correlation (0.72) with GRACE/FO data than those of Noah-MP TWS (0.57; Fig.1f). Non-seasonal CLSM TWS also exhibits smaller (1.04 cm) RMSEs than that of Noah-MP (1.16 cm). In addition to year-to-





year variation, non-seasonal GRACE/FO data exhibit multi-year variability, such as an increasing trend from 2006 to 2010 and a decreasing trend from 2013 to 2017. These interannual variations reflect combined influences of large-scale oceanic drivers such as the El Niño Southern Oscillation (ENSO) and the Indian Ocean Dipole, which strongly affect precipitation in parts of Africa (Mason & Goddard, 2001; Nicholson, 2017). Accurately capturing these climate-driven responses remains

215      challenging due to deficiencies in land surface model physics as shown here and limitations in seasonal weather forecasts (Willians et al., 2023).



**Fig.1 Domain-averaged seasonal and non-seasonal components of soil moisture of the unsaturated soil (a, b), groundwater storage (c, d) and TWS (e, f) from Noah-MP (red lines) and CLSM (blue lines) reanalysis and GRACE data (black lines). Amplitudes of mean seasonal cycles, temporal standard deviations of non-seasonal components, RMSEs and correlations between reanalysis and GRACE data are provided in matching colors of the time series.**

The strong correlation between CLSM TWS and GRACE/FO data is also contributed by secular trends in the time series which are statistically significant, at 0.014 mm/month, 0.007 mm/month, respectively. In contrast, Noah-MP TWS time series did not show statistically significant trend. Large discrepancies are observed in 2019 when GRACE/FO data show substantially lower anomalies than either model. The low anomalies in GRACE/FO data may reflect anthropogenic effects such as groundwater withdrawals during droughts which are not simulated by the models.

CLSM also better captured the seasonal amplitude of TWS changes as observed in GRACE data. Although seasonal variations simulated by both models show >0.7 correlations with GRACE/FO data, their seasonal maxima lags that of GRACE/FO data by two months, likely due to deficiencies in model physics and errors in the meteorological forcing fields. Because droughts and floods are relative to the climatological mean, evaluations in the following sections focus on non-seasonal TWS, i.e., TWS anomalies relative to monthly means.

### 3.2 Evaluation of TWS hindcasts

RMSEs of the ensemble mean TWS hindcasts of all NMME models, with respect to GRACE/FO data, exhibit distinct spatial patterns. Large RMSEs are observed in the interior western Sahel, a large region across Lake Victoria and Lake Tanganyika, and southern Zambia and Angola (Fig.2). Several factors likely contributed to these large errors. First, CHIRPS precipitation shows wetting trends in the western Sahel and the Lake Victoria and Lake Tanganyika area (Supplementary Fig.S1) which may be difficult for NMME precipitation forecasts to capture accurately, thus resulting in elevated errors. Second, in southern Zambia and Angola, strong precipitation interannual variability and the discrepancies among NMME models (Supplementary Figs.S1,2), may have contributed to inaccurate forecasts. Strong interannual variability in precipitation often leads to large errors in TWS simulation due to the challenge to accurately model long-term memory in TWS (see Fig.2 of Li et al., 2019b). Third, since the models do not simulate surface water which is detected by GRACE/FO satellites, unresolved surface water dynamics and water management activities in Lake Victoria and Tanganyika, and Lake Kariba in southern Zambia may have contributed the large errors.

CLSM forecasts show larger RMSEs than Noah-MP in Gabon, Central African Republic (CAR) and Democratic Republic of the Congo (DRC) where mean annual precipitation is among the highest, as indicated by both CHIRPS and NMME models, and where NMME models disagree considerably in interannual variability of precipitation (Supplementary Figs. S1,2). This result suggests that CLSM is more sensitive to precipitation uncertainties, due to its simulated long memory in groundwater and TWS which allow errors in precipitation and other forcing data to persist and grow over time.





250 For both models, the spatial pattern and magnitudes of RMSEs remain stable between reanalysis and hindcasts of different lead times, suggesting uncertainty in model physics, which remain the same for reanalysis and hindcast, play a strong role affecting accuracy of TWS forecasts.

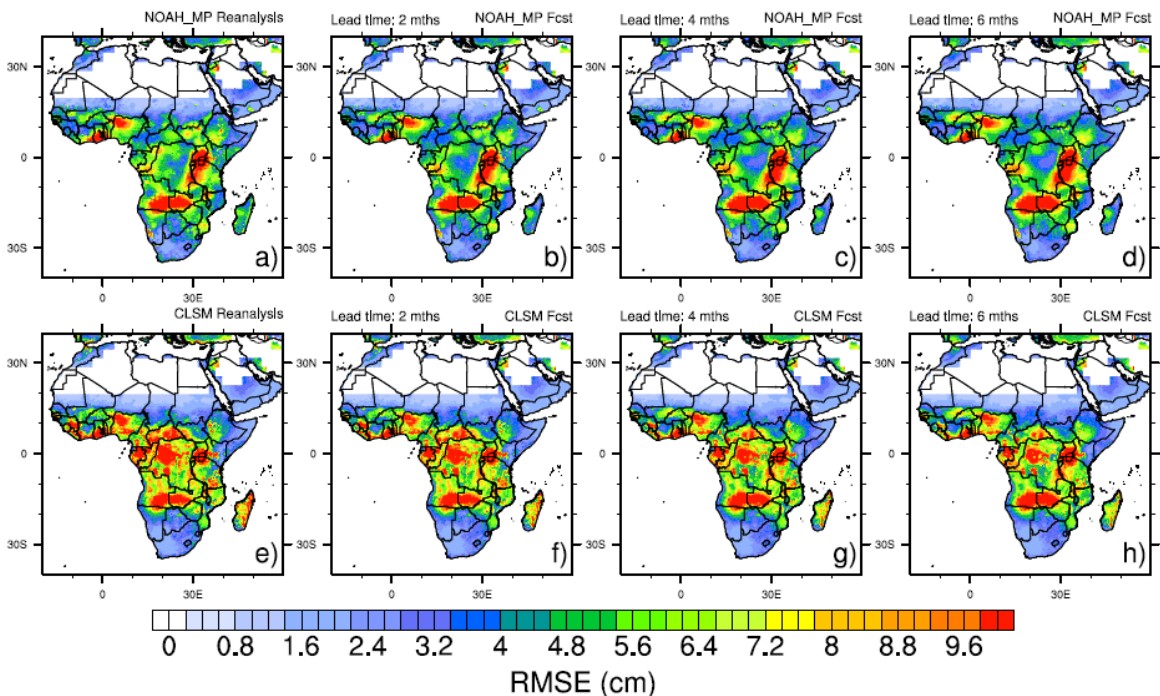

**Fig.2 RMSEs of non-seasonal reanalysis TWS, ensemble mean TWS forecasts of all NMME models with respect to GRACE/FO data**
255 **for Noah-MP (top row) and CLSM (bottom row) at three lead times.**

The correlation between the ensemble mean TWS forecasts of all NMME models and GRACE/FO data exhibits similar spatial patterns between Noah-MP and CLSM (Fig.3), suggesting precipitation forecasts likely play a strong role on correlation. However, the strength of those correlations differs notably between the two models, especially at long lead times with higher average correlation for CLSM. Correlations decrease with lead time for both models, but more rapidly for Noah-MP which, on average, decreased by 48% from the 2-
260 to 6-month lead time, compared to the 27% decrease with CLSM. Most of the deterioration in correlation is observed in central Africa where mean annual precipitation is the largest (Supplementary Fig.S1).

Similar with RMSEs, spatial patterns of correlation are consistent between reanalysis and hindcasts, indicating strong control of initial conditions on forecast skill. On average, CLSM reanalysis shows higher correlations with GRACE/FO data than that of Noah-MP, contributing to the higher forecast skill of CLSM (Fig.3).





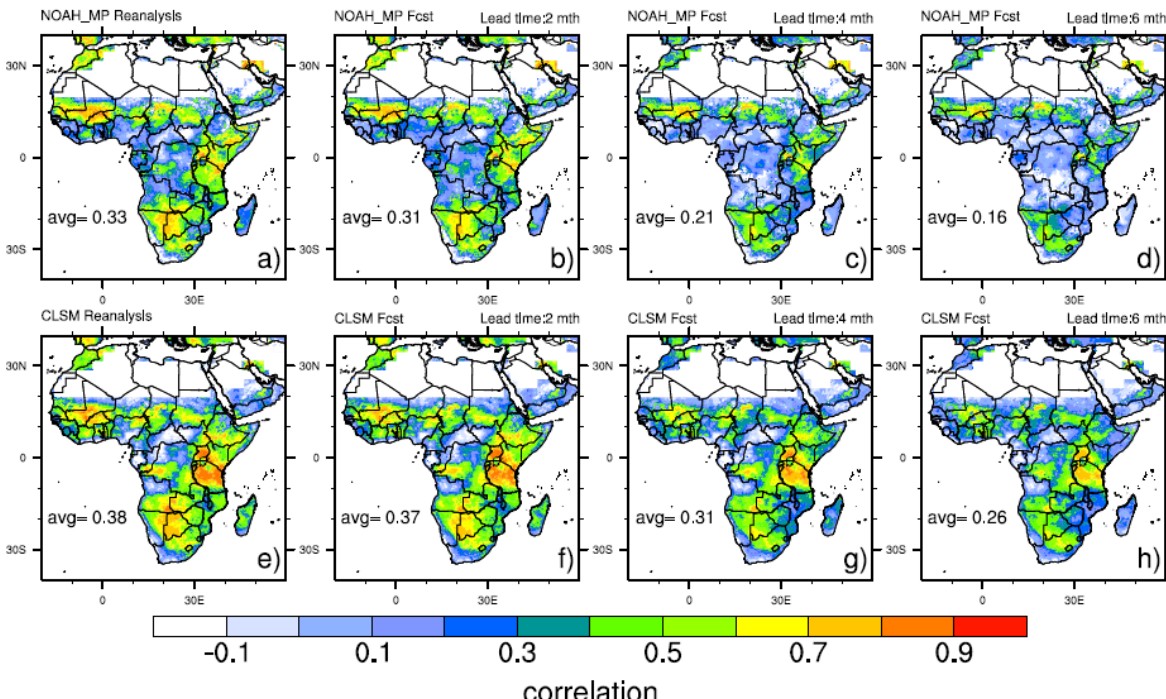

**Fig.3 Correlation between non-seasonal reanalysis TWS and ensemble mean TWS forecasts of all NMME models at three lead times, and GRACE TWS observations for Noah-MP (top row) and CLSM (bottom row). Domain average correlations are provided in inset text.**

In CAR and South Sudan, TWS forecasts from both models show near-zero correlation with GRACE/FO data likely due to the opposite trends between reanalysis TWS and GRACE/FO data (Supplementary Fig.S3). The negative trends in GRACE/FO data may reflect the impacts of deforestation, which alter the partitioning of precipitation by increasing surface runoff and decreasing soil infiltration and TWS. Although deforestation can also reduce evapotranspiration, this effect on TWS is likely minor because of reduced soil infiltration. According to the Global Forest Watch, CAR and South Sudan lost more than 20% of its primary forests during 2000-2024. Since neither Noah-MP nor CLSM accounts for land cover change, they simulated increases in TWS in response to increases in annual precipitation in that region (Supplementary Fig.S1c). Because reanalysis is used as initial conditions for each forecast issued, inaccuracy in long-term trends inevitably affected climatology and the associated anomalies.

To further explore the role of model physics and meteorological forecasts, we computed RMSEs and correlation of TWS hindcasts with respect to reanalysis (Supplementary Figs.S4,5). As expected, using reanalysis as reference results in substantially lower RMSEs and higher correlations. RMSEs show clear increases with lead time which is not obvious when evaluated against GRACE/FO data. In addition, the spatial pattern of RMSEs differs from that with respect to GRACE/FO



data. These results suggest that model physics, which are not evaluated when compared to reanalysis, have a strong impact on forecast accuracy. In contrast to evaluation relative to GRACE/FO data, CLSM forecasts show strong correlation with reanalysis in CAR across all leads due to its ability to capture interannual variability in precipitation. On the other hand, correlation for Noah-MP forecasts decreased rapidly in this region, reflecting its inability to simulate long-term TWS variability. This outcome underscores the importance of using independent data to evaluate TWS forecasts.

RMSEs of individual NMME model TWS forecasts increase with lead time for Noah-MP, reflecting growing uncertainty in meteorological forecasts (Fig.4a). In contrast, RMSEs of CLSM forecasts decrease with lead time, except those driven by GEOSv2 (Fig.4b). This behavior likely reflects CLSM's tendency to overestimate TWS interannual variability, compared to GRACE/FO data, especially in regions with pronounced precipitation interannual variability including central Africa (see Fig.2 of Li et al., 2019b). As interannual variability of NMME precipitation forecasts generally decreases with lead time (Supplementary Fig.S2), the dynamic ranges of simulated TWS are suppressed, leading to reduced RMSEs. This explains why largest RMSEs are observed in hindcasts by GEOSv2 which also increase with lead time, owing to its largest precipitation interannual variability (Supplementary Fig.S2). These results again suggest that model physics may have stronger influences on accuracy of TWS forecasts than meteorological forecasts.



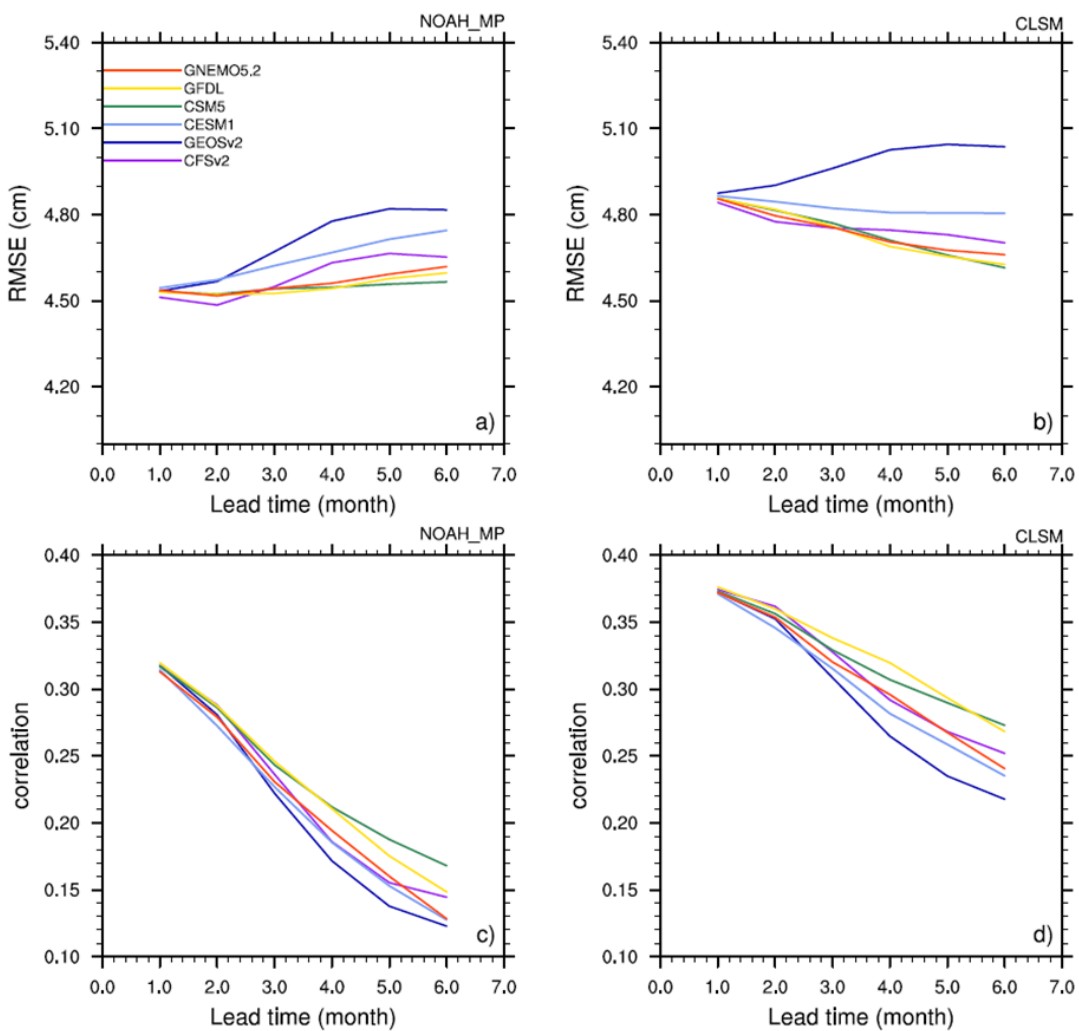

**Fig.4 Average RMSEs (top row) and correlations (bottom row) of ensemble mean TWS forecasts of individual NMME models relative to GRACE/FO data for Noah-MP (left column) and CLSM (right column) as a function of lead time.**

Correlations between ensemble mean TWS hindcasts of individual NMME models and GRACE/FO data follow the similar pattern as those of all-model ensemble mean: correlations decrease with lead time (Figs.4c,d). In addition, CLSM forecasts, on average, exhibit higher correlation than those of Noah-MP at all lead times.

Among all NMME models, GFDL and CSM5 produce the most accurate forecasts, with the lowest RMSEs and the highest correlations, whereas GEOSv2 produced the least accurate TWS forecasts, yielding the largest RMSEs and lowest correlations

for both Noah-MP and CLSM at all lead times. In addition to strong interannual variability of GEOSv2 discussed above,





previous studies showed that GEOS precipitation forecasts are less consistent among its ensemble members than other NMME models (Becker et al., 2014), indicating larger uncertainty in GEOS precipitation forecasts.

ROC scores for the lower tercile forecast exhibit similar spatial patterns as correlations (Fig.5). CLSM shows considerable skills, achieving >0.6 ROC scores over 50% of the domain across all lead times. The 0.6 threshold for predictive skill is based

on the guideline by the Met Office (Met Office). In contrast, Noah-MP forecasts exhibit lower ROC scores overall and the scores decrease quickly with increases in lead time, especially in central Africa, with only 35% of areas achieving >0.6 ROC scores at the 6-month lead time. Both models perform well in the Sahel (minus the northern edge), the Horn of Africa, and the eastern part of southern Africa where trends in reanalysis TWS generally agree with those of GRACE/FO data (Supplementary Fig.S3). As expected, both models scored low ROC values in CAR and South Sudan, due to the opposite TWS trends with

GRACE/FO data as discussed previously.

ROC scores for the upper tercile forecast show similar spatial patterns but with slightly higher values than those of the lower tercile (Supplementary Fig.S6). This difference likely reflects the uneven occurrences of wet and dry anomalies over the relatively short study period (2003-2020).

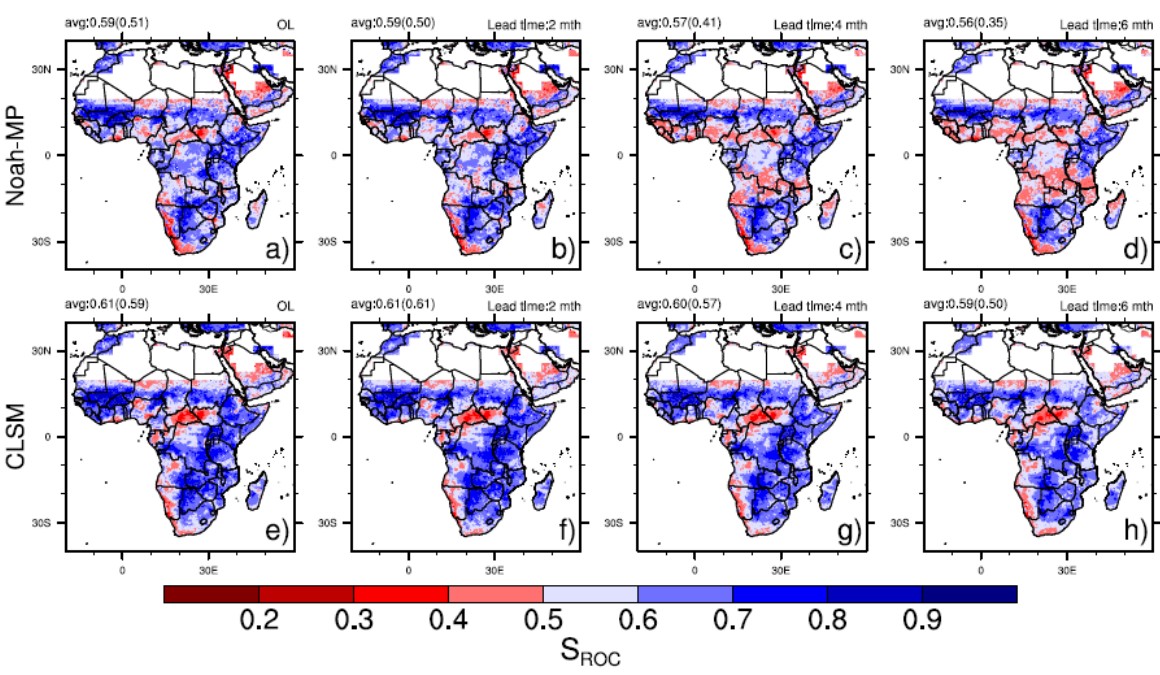




**Fig.5 ROC scores (S_roc) of lower terciles of ensemble mean TWS forecasts of all NMME models by Noah-MP (top row) and CLSM (bottom row) with respect to GRACE/FO data. The upper left text indicates average S_roc and the fraction of area (in parentheses) with S_roc>0.6.**

### 3.3 Persistence of TWS processes

Persistence is known to help enhance hydrological prediction skill. To properly examine persistence, we computed the autocorrelation of TWS time series from the two re-analyses and GRACE/FO data at three lags (Fig.6). CLSM reanalysis TWS exhibits higher autocorrelations than Noah-MP across central Africa at all three lags, while Noah-MP simulates strong persistence in the drier northern and southern Africa. GRACE/FO data reveals a different spatial pattern in persistence, with strong persistence in the interior Sahel and a large swath area across Lake Victoria, and Zambia and southern Angola.

Regardless of data sources, strong persistence in TWS is generally associated with strong long-term trends in TWS (Supplementary Fig.S3). The positive trends in the western Sahel shown in GRACE/FO data have been linked to the northward shift of the northern African monsoon, which has led to increases in wet extremes in the region (Monerie et al., 2021; Rodell and Li, 2023). The increasing trend was better captured by the Noah-MP reanalysis than CLSM. On the other hand, CLSM reanalysis better reflected the wetting trends in the Lake Victoria region observed in GRACE/FO data. Note that while surface water is a major TWS component in Lake Victoria, the increasing trend in GRACE/FO data mainly reflected precipitation surpluses around 2020, not due to water management activities (Boergens et al., 2024). As discussed earlier, the strong persistence in CAR for CLSM reanalysis is associated with increases in precipitation.





**Fig.6 Autocorrelation of TWS time series for Noah-MP (top row) and CLSM (middle row) reanalysis, and GRACE/FO data (bottom row) at three lags. Upper right text indicates average autocorrelation and fraction of area with autocorrelation>0.37 (in parentheses).**


On average, CLSM reanalysis shows stronger persistence than Noah-MP, with a larger fraction of the domain exhibiting >0.37

auto-correlation (representing e-folding time) at the 2- and 4-month lags.  Compared to reanalysis, GRACE/FO data shows



lower persistence at the 2-month lag, but substantially higher autocorrelation at 4- and 6-month lags. The area with >0.37 autocorrelation in GRACE/FO data remains high, above 75% even at the 6-month lag.

To further explore contributing factors to TWS persistence, we examine autocorrelation of soil moisture of the unsaturated soil and groundwater storage from the reanalysis (Fig.7). It is clear that the weak persistence in Noah-MP reanalysis TWS is closely linked to the weak persistence in its simulated groundwater. At the 1- and 2-month lags, persistence in Noah-MP groundwater is lower than that of its soil moisture. In contrast, CLSM groundwater shows much stronger persistence than its simulated soil moisture, with average persistence in groundwater nearly identical to that of TWS. Compared to reanalysis, GRACE/FO observations exhibit a more rapid decline in persistence at 0–2-month lags, but a slower decline afterwards.

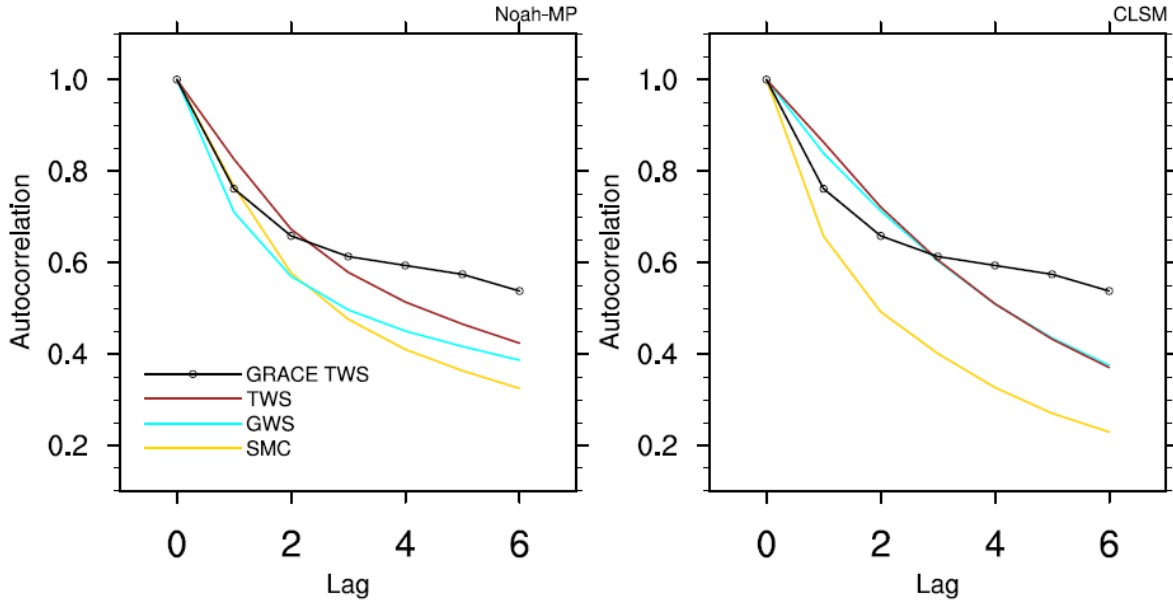

**Fig.7 Domain average autocorrelation for soil moisture of the unsaturated soil, groundwater storage and TWS.**

### 3.4 TWS forecast percentiles

To explore the value of TWS forecasts, we examine TWS percentile maps derived from CLSM forecasts initialized in December 2015 (Fig.8). Severe droughts (<10th percentile) affected much of southern Africa throughout the 6-month forecast period (Fig.8), in association with the 2015-2016 El Niño event which typically brings dry conditions to southern Africa (Mason & Goddard, 2001). Drought conditions were further intensified by record-setting global temperature in 2016. TWS percentiles indicate most severe droughts (<5th percentile) between December 2015 and March 2016. In particular, the extent and severity of drought conditions forecasted for March 2016 are generally consistent with the FEWS NET drought assessment





released in March 2016 which is based on cumulative precipitation analysis (FEWS NET, 2016). Additionally, this drought
caused up to 66% decline in crop production and affected at least 18 million people (Ainembabazi et al., 2018). This highlights
the potential and benefit of TWS forecasts for providing early warnings for severe and persistent droughts. Drought conditions
are also evident in the southwestern coastal countries including Ghana and Gabon.

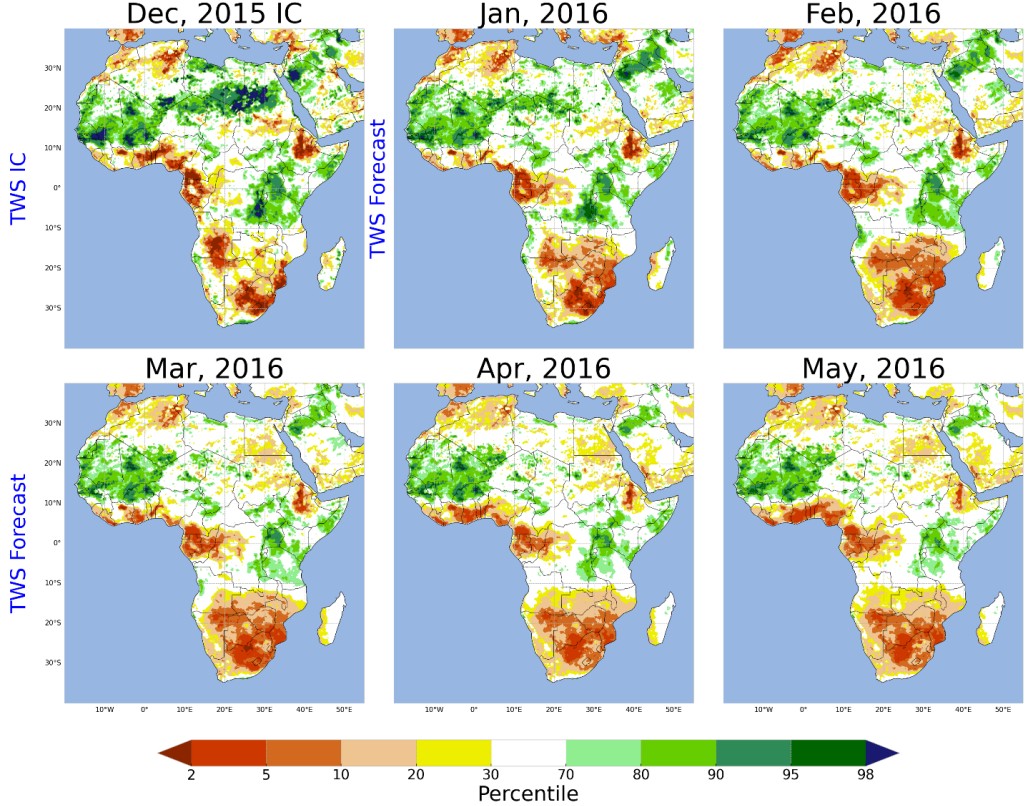

**Fig.8 TWS percentile maps derived from CLSM TWS forecasts of all NMME ensemble models, initialized in December 2015.**


Wetter conditions are observed in eastern Africa, especially in Rwanda where relentless rainfalls in May 2016 trigged
landslides that killed dozens of people and destroyed hundreds of homes (Bishumba, 2016). Above normal conditions also
occurred in the northern part of western Africa such as Mauritania, Mali and Senegal from December to May. These wet
conditions may have contributed to the devasting floods in Mali and Burkina Faso in July 2016 (FloodList, 2016)

Probability maps indicate strong agreement among NMME forecasts for these wet and dry anomalies (Supplementary Fig.S7).
In addition, probabilities generally decrease with increases in lead time, reflecting increased uncertainty in both
meteorological and TWS forecasts.



Surface and root zone soil moisture forecasts show similar dry and wet patterns as TWS forecasts (Supplementary Figs.S8,9). However, the severity and extent of wet and dry anomalies may differ depending on the indicator. For instance, surface soil moisture identified <5$^{th}$ percentile drought conditions in Gabon in December 2015 which improved to 10-20$^{th}$ percentiles by May 2016, while TWS anomalies remained in <10$^{th}$ percentiles throughout the 6-month period. Similarly, the wet anomalies in eastern Africa returned to normal more quickly in surface soil moisture than in TWS. Root zone soil moisture percentiles also show faster changes in anomalies with lead time than TWS percentiles, but less so than surface soil moisture (Supplementary Fig.S9). These results reflect lagged responses to changes in precipitation as the soil depth increases.

**4 Summary and discussions**

We evaluated terrestrial water storage (TWS) forecasts produced by the FLDAS Hydrological Forecasting System (FLDAS-Forecast) in Africa using GRACE/FO TWS observations. Statistical analyses indicate that the Catchment land surface model (CLSM) demonstrates considerable skills in forecasting terciles at S2S scales, with >0.6 ROC scores (the threshold indicating predictive skill) over more than 50% of the study area across 1- to 6-month lead times. CLSM forecasts also show stronger correlations with GRACE/FO data than those of Noah-MP, especially at long lead times. A key contributing factor is that CLSM reanalysis, used as initial conditions, better captured the interannual variability in GRACE/FO observations, with >0.7 correlation for domain averaged TWS anomalies. Furthermore, the skill in initial conditions is retained at longer lead times through persistence associated with strong interannual variability in its simulated TWS. As interannual variability determines climatology, the combination of more accurate initial conditions and persistence led to more accurate anomaly forecasts.

In contrast, Noah-MP forecasts showed low skills, especially in central Africa where ROC scores generally fall below 0.6 at the 4- and 6-mth lead times. This reduced performance is partly attributed to the reanalysis-based initial conditions that exhibit smaller interannual variability compared to GRACE/FO data, degrading accuracy of climatology and anomalies. Weak interannual variability also led to a more rapid decline in forecast skill with lead time in Noah-MP.

Accuracy of TWS forecasts showed strong dependency on interannual variability of precipitation forecasts. TWS forecasts based on GEOSv2 precipitation, which exhibits the largest interannual variability, showed the lowest correlation and highest RMSEs with respect to GRACE/FO observations. On the other hand, TWS forecasts based on GFDL and CSM5 precipitation, which have the lowest interannual variability, yielded the highest correlations and lowest RMSEs. Predicting precipitation interannual variability is challenging due to limited temporal samples (McKinnon and Deser, 2021) and can therefore entail greater uncertainty. On the other hand, model physics can exert stronger influences on accuracy of TWS forecasts than precipitation forecasts. This effect is more evident with CLSM which may overestimate TWS responses to precipitation interannual variability and thus, can produce more accurate TWS forecasts with precipitation forecasts of reduced interannual variability such as for some NMME models at long lead times.





Statistical analysis showed that forecast skill is substantially lower when evaluated against GRACE/FO data than when assessed relative to reanalysis for both models, further underscoring substantial uncertainty in model physics. These effects are most pronounced in central Africa where forecast skill varies considerably between the two models. As discussed above, the ability to simulate multi-year responses to precipitation variability is critical for skillful TWS forecast in regions with strong precipitation interannual variability. CLSM performs better in this regard because it realistically represents upward

groundwater movement during extended dry periods, sustaining ET and creating storage capacity for groundwater recharge when precipitation returns, thereby resulting in pronounced interannual variability in TWS. In contrast, Noah-MP's weak representation of capillary rise limits groundwater storage capacity and its ability to capture long-term precipitation variability, especially during prolonged droughts.

Auto-correlation analysis showed that TWS persistence is strongly impacted by persistence in simulated groundwater,

underscoring the need to improve groundwater representation, especially in capturing long-term precipitation variability. In addition, this study shows that while persistence contributes to predictability, inaccurate persistence can lead to erroneous forecasts. This is especially true in regions experiencing secular trends in precipitation and anthropogenic effects such as groundwater withdrawals and deforestation that are often poorly simulated by land surface models (Döll et al., 2014).

As discussed above, interannual variability is critical for accurately forecasting TWS anomalies. GRACE/FO data assimilation

has been shown to be an effective means to constrain the temporal variability of TWS and its components (Li et al., 2019b). Due to the computational cost and 2-4 months of data latency, GRACE/FO data assimilation is only feasible for reanalysis-based simulation, not suited for forecast runs. As a result, while GRACE/FO data assimilation may improve reanalysis-based initial conditions, deficiencies in model physics such as inability to properly simulate responses to precipitation variability can prevent improvements in initial conditions from translating into enhanced forecast skill at longer lead times. Therefore,

improving physical representation of TWS processes, especially their responses to anthropogenic effects and long-term precipitation variability, should be a key priority for future research aimed at improving TWS forecasts.

**Data availability:** FLDAS forecast data are available at https://ldas.gsfc.nasa.gov/fldas/models/forecast.

**Author contribution**: BL: conceptualization, formal analysis, methodology, investigation, visualization, writing (original

draft preparation). AH: Data curation, software, writing (review and editing). AM: project administration, supervision, writing (review and editing). KS: project administration, writing (review and editing). SS: Funding acquisition, supervision, writing (review and editing) WA: writing (review and editing).

**Competing interests**: The authors declare that they have no conflict of interest.

**Acknowledgement**: This work was supported by NASA grant # 80NSSC23K0559 entitled "Improving a process-based

understanding of how terrestrial water storage can improve S2S hydrologic forecasts skill in data-sparse regions." A. McNally,



K. Slinski, A. Hazra, B. Li, and W. Anderson were also supported by the Bureau of Humanitarian Assistance, U.S. Agency for International Development, under the terms of PAPA BHA22H00005 "FAMINE EARLY WARNING SYSTEMS NETWORK (FEWS NET)". The opinions expressed in this paper are those of the authors and do not necessarily reflect the views of the U.S. Agency for International Development. K. Slinski was also supported by NASA grant # 80NSSC23M0032 entitled
"NASA Harvest: NASA's Global Food Security and Agriculture Consortium."

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
