# Peer review of "Skills in sub-seasonal to seasonal terrestrial water storage forecasting: insights from the FEWS NET land data assimilation system"

_EGUsphere, 2025_

## Author Comment (AC1)

**Review#2**

**Summary:**

The authors evaluate the skill of FEWS NET S2S terrestrial water storage (TWS) hindcasts over Africa. This study is relevant for improving forecasts and the management of extreme conditions across the continent, contributing to enhanced climate resilience. The focus lies on the differing performance of two land surface models—CLSM and Noah-MP—in comparison with GRACE observations. The analysis also includes the decomposition of TWS into its individual storage components and an assessment of TWS persistence. Evaluation metrics used are RMSE, correlation, and ROC.

Overall, the insights presented in the paper are valuable and merit publication. However, three major points should be addressed:

- Some parts of the manuscript are difficult to follow. This can likely be resolved by adding a few clarifying explanations; specific suggestions are provided in the detailed comments.
- The manuscript contains contradictory statements regarding the role of surface water bodies in Africa. In the data section, they are described as small, yet later results indicate that they are relevant. Given that surface water bodies can substantially influence TWS variability in many African regions—and considering the existing literature on this topic—it may be advisable to remove their contribution from the GRACE data before performing the analysis.
- The evaluations in Section 3.1 are based on time series averaged over the entire African continent. In my view, such continental-scale averages offer limited interpretive value due to the large diversity of climate zones and hydrological regimes. The authors also appear to have difficulties interpreting the signal they observe. It would be more informative to present time series for a selection of representative regions.

Thank you for your supportive comments. We agree some of the statements are difficult to follow and have revised the Results section extensively to better explain the purpose and result of each analysis.

Regarding surface water, we agree that removing surface water storage from GRACE data can help improve the diagnosis of model physics issues in areas with large surface water bodies. However, due to the unique satellite design and data processing processes, signals of surface water are inevitably spread to a large area surrounding a surface water body and thus, simple water balance calculation, i.e., subtracting surface water storage from GRACE data, at the grid cells is insufficient to remove surface water signals from the GRACE record. Properly separating surface water signals from GRACE observations requires specialized techniques and expertise (Deggim et al., 2021; Sharifi et al., 2025), which are beyond the scope of this study. In addition, accurately computing surface water storage changes requires careful determination of lake shore areas (Boergens et al., 2024), routing algorithms and modeling surface water and groundwater interaction, all of which are subject to large uncertainties themselves (Bierkens and Wada 2019). There is also substantial uncertainty in modeling soil moisture and groundwater, as

shown in this study. Therefore, we believe this issue is more appropriately addressed in a dedicated study to disentangle contributions of individual TWS components. We also do not expect this issue to substantially affect the outcome of this study given that lake areas only constitute a small fraction of the continent.

We acknowledge the importance of isolating surface water signals and highlight the need for future research in the final section as the following," *Separating surface water signals from GRACE/FO data can improve the diagnosis of modeled TWS processes in regions dominated by large surface water bodies. However, due to the unique satellite design and data processing of GRACE/FO, surface water storage changes are spatially spread over broad areas surrounding surface water bodies. As a result, simple grid-scale mass balance calculations are insufficient, and specialized modeling approaches and techniques are required to properly isolate surface water contributions to GRACE/FO TWS observations (Deggim et al., 2021; Sharifi et al., 2025). Because such techniques and expertise are not widely available, there is a need for surface water datasets that are harmonized and scaled to GRACE/FO observations to facilitate the separation of surface water signals. Beyond temporally consistent local observations of surface water (water elevation and extent), this effort requires explicitly modeling surface water and groundwater interactions, currently absent in most land surface models, to quantify contributions of groundwater to surface water storage changes which can be substantial in wet climates (Bierkens and Wada 2019)*".

Bierkens, M. F. and Wada, Y.: Non-renewable groundwater use and groundwater depletion: a review, Environ. Res. Lett., 14, 063002, https://doi.org/10.1088/1748-9326/ab1a5f, 2019.

Deggim, S., Eicker, A., Schawohl, L., Gerdener, H., Schulze, K., Engels, O., Kusche, J., Saraswati, A. T., van Dam, T., Ellenbeck, L., Dettmering, D., Schwatke, C., Mayr, S., Klein, I., and Longuevergne, L.: RECOG RL01: correcting GRACE total water storage estimates for global lakes/reservoirs and earthquakes, Earth Syst. Sci. Data, 13, 2227–2244, https://doi.org/10.5194/essd-13-2227-2021, 2021.

Sharifi, E., Haas, J., Boergens, E., Dobslaw, H., and Güntner, A.: Technical note: GRACE-compatible filtering of water storage data sets via spatial autocorrelation analysis, Hydrol. Earth Syst. Sci., 29, 6985–6998, https://doi.org/10.5194/hess-29-6985-2025, 2025.

Please see our response to your third comment below.

**Specific comments:**

**Abstract:**

- l. 18: "> 0.6 relative operating characteristics" →this measure is not clearly defined. Do you think that it is obvious what it represents?

We revised this sentence to say, "*Results show that CLSM provides more skillful TWS forecasts than Noah-MP, with relative operating characteristics (ROC) scores exceeding 0.6*

*(the threshold for predictive skill) for tercile forecast over more than half of the study domain across the 1-6 months lead times".*

We also revised section 2.6 to provide more background information about ROC like this, "*Additionally, skill in forecasting terciles is assessed using the relative operating characteristic (ROC) score, a commonly used evaluation metric measuring the ratio of hit rates to false alarm rates (Met Office). A ROC score of 1 indicates a perfect forecast. ROC scores below 0.5 suggest no skill, while scores above 0.6 indicate predictive skill (Met Office). High ROC scores and strong correlation are commonly interpreted as indication of skillful forecasts (e.g., Yuan and Zhu, 2018)*".

Before discussing ROC results in section 3.2, we also added "*While RMSEs and correlation quantify the magnitude of discrepancies and the temporal consistency between two time series, they do not directly assess the ability to accurately forecast wetter and drier conditions. Therefore, we use ROC scores to evaluate the performance of Noah-MP and CLSM in predicting terciles, corresponding to below-normal, near-normal and above-normal conditions*".

- L 27: you talk about multi-year variability, but before you said that you do S2S forecasts. This is confusing… so what is the aim here?

Yes. It was confusing. We have clarified this by adding these sentences, "*Accurate representation of interannual variability is essential for S2S forecasts because TWS, as a long memory process, can carry wet and dry information over months, and interannual variability also directly affects climatology used to determine anomalies*".

- L 32: "relative to reanalysis" → which reanalysis are you referring to?

This sentence has been removed from Abstract.

- L.35: Would it make sense to explain the term "persistence"? Maybe make clearer using the term "temporal persistence".

Thanks, we revised it to say temporal persistence.

**1 Introduction:**

- L 56/57: I could not understand what you want to say with this sentence.

This sentence has been revised to say, "*More importantly, evaluations using simulated TWS as reference masks the impact of model physics and thus are unable to assess uncertainties in model physics and even mischaracterize persistence and its role in TWS forecast*".

- L 60: I would say that groundwater being a potential source of predictability for TWS depends mainly on its variability?

*This sentence has been revised to "*Groundwater, located in the deeper subsurface, has longer memory than other near surface processes such as soil moisture and its long-term temporal variability may contribute to TWS predictability*".*

- L 64: the sensitivity to biases in the meteorological forcing depends much on the response time, which might be much longer than the S2S scale.

  *We replaced "biases" with "errors" to reflect all aspects of precipitation uncertainties. In addition, longer scale variabilities can still affect S2S forecasts due to their impact on initial conditions and climatology as we explained above.*

- L 76: you may add the following two studies to your discussion (if you think it is fitting!). They assimilated GRACE-based forecasts of TWS into hydrological models in order to improve the forecast skill of the models:
    - Li, F., Springer, A., Kusche, J., Gutknecht, B., Ewerdwalbesloh, Y. (2025). Reanalysis and Forecasting of Total Water Storage and Hydrological States by Combining Machine Learning With CLM Model Simulations and GRACE Data Assimilation. Water Resources Research, e2024WR037926, https://doi.org/10.1029/2024WR037926
    - Li, F., Kusche, J., Sneeuw, N., Siebert, S., Gerdener, H., Wang, Z., ... & Tian, K. (2024). Forecasting next year's global land water storage using GRACE data. Geophysical Research Letters, 51(17), e2024GL109101. https://doi.org/10.1029/2024GL109101

*Thanks for bringing our attention to these papers. We added this sentence to the Introduction: "*In recent years, the record has also been leveraged to train machine learning models for forecasting TWS (e.g., F. Li et al., 2024 & 2025)*".*

- L 90: Here you talk about hindcast for the first time. Before you only talk about forecast. You may introduce this.

*Thanks for the suggestion. We revise this paragraph to read as follows: "*The primary goal of this study is to provide an objective evaluation of the skill of S2S TWS forecasts from FLDAS-Forecast using GRACE/FO observations. To this end, we analyze TWS hindcasts for the historical period 2003-2020. The hindcasts were generated using the same set of NMME models employed in the operational FLDAS forecasts (2011-present), except that two of the NMME models used a reduced number of ensemble members (Hazra et al., 2023). Initial conditions for the hindcasts are derived from model simulations forced by reanalysis-based meteorological datasets. Consequently, TWS hindcast skill reflects the combined influence of land surface model physics, meteorological hindcasts, and the reanalysis forcing used to generate initial conditions.*

- L 91: multi-model → are there more models involved besides CLSM and Noah?

*There are only two models which can still be considered as multi-model. However, we revised this sentence to:" *By leveraging the multi-model framework of FLDAS-Forecast,*

*including two land surface models and a full ensemble of NMME meteorological forecast models, the evaluation aims to improve understanding of how model physics employed by land surface models influence TWS forecast skill and how they interact with meteorological forecasts*".

- L. 95/96: please provide references for the past studies

References have been added: "*Unlike past studies where S2S forecasts were evaluated by season when they were issued (e.g., Shukla et al., 2019; Hazra et al., 2023)*",

- L. 97: It is not clear to me how autocorrelation analysis can be applied to processes. I would say it can be applied to variables.

"Processes" has been replaced by "*simulated TWS processes and GRACE/FO observations*".

**2 Data and evaluation metrics**

- L101: how do you make sure that CHIRPS based precipitation and the other fields are consistent? And why do you use different fields for generating the initial conditions than used afterwards used for the hindcasts.

We now explain that CHIRPS and MERRA-2 are interpolated to a common grid for model simulation.

 CHIRPS and MERRA-2 cannot predict the future and therefore, to properly evaluate forecast skill, hindcasts must be generated using the same forcing data used for forecasting, the NMME suite. As noted above, hindcasts and forecasts are nearly identical, differing only in their time period.

- L. 147: on the long-term I agree that groundwater variability is balanced by P and ET, but at S2S scale I have doubts. The papers you cite both refer to the US, which has very different climate regimes and soils. Do you have evidence over Africa that groundwater is influenced by ET on S2S scale?

We state groundwater temporal variability is controlled by precipitation and ET, not balanced out by P and ET which implies net-zero groundwater storage changes. In unconfined aquifers, groundwater temporal variability is a result of combined influences of P and ET based on the cited publications. We added Li et al., 2019b & Ascott et al., 2020 that employ in situ groundwater in Africa. In addition, when describing the models in section 2.3, we added, "*In addition, both models do not simulate water storage changes in confined aquifers which are also detected by GRACE/FO satellites*".

- L. 154: It was shown by Ndehedehe et al (2017) that lake Volta contributes to 40% of TWS trend in the Volta basin. So I think that in particular over West Africa you cannot neglect surface water bodies when comparing to GRACE.

- o    Christopher E. Ndehedehe, Joseph L. Awange, Michael Kuhn, Nathan O. Agutu, Yoichi Fukuda, Analysis of hydrological variability over the Volta river basin using in-situ data and satellite observations, Journal of Hydrology: Regional Studies, Volume 12, 2017,Pages 88-110, ISSN 2214-5818, https://doi.org/10.1016/j.ejrh.2017.04.005.

We agree and that's why we say, "surface water contribution to TWS is generally smaller compared to other TWS components, except in areas with large surface water bodies" in Line 154.  We revised this paragraph to "*Neither model simulates surface water, which is detected by GRACE/FO satellites.  However, surface water contribution to TWS is generally smaller compared to other TWS components, except in areas with large surface water bodies (Rodell & Famiglietti, 2001; Getirana et al., 2017; Deggim et al., 2021).  The implications of neglecting surface water storage for the evaluation metrics are discussed in section 3*".

We also revised the beginning paragraph of section 3.2 to read like this: "*RMSEs of the ensemble mean TWS hindcasts of all NMME models, with respect to GRACE/FO data, exhibit distinct spatial patterns (Fig.3).  Large RMSEs are observed in the interior western Sahel, a large region across Lake Victoria, Lake Tanganyika, and Lake Volta as well as southern Zambia and Angola, for both models.  As the models do not simulate surface water which is detected by GRACE/FO satellites, unresolved surface water dynamics and water management activities may have contributed to errors in lake areas.  In addition, uncertainties in precipitation forcing data, from both reanalysis and hindcasts, especially under a changing climate, may further amplify errors in simulated TWS.  As discussed earlier, the East African Rift, which includes Lake Victoria, has seen increased precipitation variability (Boergens et al., 2024); similarly, Southern Africa including southern Angola has been experiencing erratic precipitation patterns and more severe meteorological droughts in recent years (Trisos et al., 2022; Correia et al., 2025).  However, considering that the reanalysis exhibits similar spatial patterns and magnitudes of RMSEs as the hindcasts (Figs.3a,e), deficiencies in model physics are likely the dominant contributor to RMSEs in TWS hindcasts*".

- L 161: Could you clarify: do you evaluate in the following the ensemble mean?

Yes.  We added this sentence here (the end of section 2.3)," *In section 3, we evaluate both the ensemble mean TWS hindcasts of individual NMME models and the all-model ensemble mean hindcasts*".

- L 170: why do you remove only 5 years as temporal mean? What about regions that have strong interannual variability? I would expect that in particular the percentile maps shown in Fig. 8 can be significantly affected by the choice of the time span for the temproal mean.

We didn't remove the 5 years mean from GRACE data which are provided as anomalies relative to the 2004-2009 temporal mean.  To be consistent with GRACE data, we removed the 2004-2009 mean from modeled data (hindcasts and reanalysis).  We further removed the

monthly climatology derived from the entire evaluation period (2003-2020) at each grid cell which should take care of the interannual variability issue.

This is how we explain the data processing procedure in section 2.5: "*To ensure consistency with GRACE/FO data, we first removed the temporal mean of simulated TWS for 2004-2009 at each grid cell to align the model's mean period with that of GRACE/FO. We then computed non-seasonal TWS anomalies by subtracting the monthly mean (climatology), one for each calendar month, from the simulated TWS and GRACE/FO time series for their overlapping period, 2003-2020. Unless otherwise noticed, all results presented in section 3 are based on the non-seasonal TWS anomalies*".

- L. 178: do you mean that you removed the climatology?

Yes. The sentence has been revised to say monthly climatology.

- L. 180: you could explain at some point the differences between reanalysis and hincasts.

We revised the last paragraph of section 2.3 to clearly define reanalysis and hindcast. The section title has also been updated to include reanalysis. This is the revised text: "*Model simulations were first performed by driving Noah-MP and CLSM with CHIRPS precipitation and non-precipitation fields from MERRA-2. Because MERRA-2 is a reanalysis product, these simulations are collectively referred to as the reanalysis. TWS hindcasts with lead times of 1 to 6 months were then generated by forcing Noah-MP and CLSM with the NMME hindcasts described above, using the corresponding reanalysis output as initial conditions for each hindcast. Since CHIRPS and MERRA-2 are constrained by hydrological and atmospheric observations, initializing the hindcasts with the reanalysis, rather than modeled states driven solely by NMME meteorological hindcasts, helps reduce uncertainty in TWS hindcasts*".

- L. 185: you used white color in the figures to highlight regions that were masked out. This is not clear, better use gray color.

Thanks for pointing this out. Those regions are now masked using the grey color.

- L. 186: you could make the percentiles clearer by an equation?

An equation has been added in section 2.5 to show how percentiles are calculated.

**3 Results**

- L 197 ff: The structure is difficult to understand. You introduce the three influencing factors, and then say how each of them is isolated. However, for the first to factors I cannot understand how they are connected to what you analyze. Could you make this more clear? How are initial conditions connected to temporal variability of reanalysis? How are meteorological forecasts connected to different lead times? And to isolate model physics shouldn't you compare to GRACE?

Indeed, this paragraph was confusing. We have revised it extensively to read like this: "*The skill of TWS hindcasts is influenced by several factors, including initial conditions, meteorological hindcasts, and the underlying land surface model physics which affect both the reanalysis-based initial conditions and TWS hindcasts. Because these influences are interrelated, fully isolating their individual contribution to hindcast skill is inherently challenging. To address this, we conduct a series of complementary evaluations using both GRACE/FO data and the reanalysis as reference.*

*We begin by examining the temporal variability of reanalysis soil moisture and groundwater, which are used as initial conditions for TWS hindcasts, to assess their relative contribution to temporal variability and accuracy of TWS (section 3.1). We then evaluate TWS hindcasts and the corresponding reanalysis using GRACE/FO observations to quantify forecast skill for each land surface model and NMME forcing model, and accuracy of initial conditions (section 3.2). Since TWS hindcasts differ from the reanalysis only in their meteorological forcing fields, evaluating TWS hindcasts using the reanalysis as reference helps isolate uncertainties in NMME hindcasts. In addition, because model physics are effectively masked when using reanalysis as reference, differences between the two sets of evaluation metrics, one relative to GRACE/FO and another relative to the reanalysis, indicate impacts of model physics employed by an individual land surface model (section 3.3)*".

Comparing to GRACE data does not isolate model physics because simulated TWS is also influenced by uncertainties in forcing data.

- L 203: Are you sure that it makes sense to average over such a huge study area? Wouldn't it be more appropriate to look at regions of interest? For instance, you are not able to interpret the interesting signals that you highlight in L 211, because you do not know from which part of your region they come.

In response to your comment, we added a new analysis (Fig.2 & Table 2) to show basin-scale TWS comparisons in section 3.1.

[Figure]

*Fig.2 Average seasonal (left column) and non-seasonal (right column) components of TWS from the Noah-MP (red lines) and CLSM (blue lines) reanalysis and GRACE/FO data (black lines) for the six largest river basins in Africa (Basin delineations are shown in Supplementary Fig.S1a). RMSEs and correlations between the reanalysis and GRACE data are shown in matching colors of the corresponding time series.*

Multiyear variability is now discussed with respect to these basin averaged TWS time series: "*To further evaluate the performance of reanalysis TWS, we analyze temporal variability of basin-average TWS time series for the six largest river basins in Africa (see Supplementary Fig.S1a for basin delineations). Seasonal TWS for both models show high correlations (generally >0.7) with GRACE/FO data in most basins, indicating that the timing of the seasonal cycle is well captured by the model (Fig.2, left column). Noah-MP often exhibits slightly higher seasonal correlations with GRACE/FO than CLSM (e.g., Congo, Niger, Nile, and Chad); however, its performance degrades substantially in*

*Orange, where the correlation is near zero (Fig.2g).  This low correlation is attributed to the misalignment in the timing of annual minimum TWS, with Noah-MP reaching its seasonal low in February, whereas GRACE/FO observations (and CLSM) reach their seasonal low in November.*

*In contrast, the non-seasonal component of reanalysis TWS exhibits notably lower correlations with GRACE/FO (Fig.2, right column), reflecting greater challenges in simulating interannual TWS variability. CLSM generally achieves higher correlations than Noah-MP in the central and northcentral basins (Congo, Nile and Zambezi), whereas Noah-MP performs better in the northwestern basins (Niger and Chad). RMSEs are lower for each model in three of the six basins.  Consistent with the domain averaged analysis (Fig.1), CLSM simulates larger seasonal variability and stronger interannual variability than Noah-MP, both of which are in closer agreement with GRACE/FO data (Table 2).*

*Non-seasonal TWS often exhibits strong interannual variability driven by both climate variability and anthropogenic effects.  For instance, TWS reached maxima in the Zambezi and Orange basins in 2011 in association with the La Niña event (Figs.2h,j), which typically brings wetter conditions to southern Africa (Mason and Goddard, 2001; Scanlon et al., 2022;).  Similarly, the strong TWS increases in the Nile basin in 2019, evident in GRACE/FO data (Fig.2f), is linked to a strong positive phase of the Indian Ocean Dipole (Scanlon et al., 2022) and enhanced precipitation variability in the Eastern African Rift (Boergens et al., 2024).  In the Niger basin (Fig.2d), GRACE/FO shows a strong persistent increasing trend (0.048 cm month-1) that has been linked to conversion of shrubs to crops (Favreau et al., 2009) and corroborated by well data (Scanlon et al., 2022).  Since the models do not represent land cover change, the reanalysis TWS exhibits smaller trends (0.019 cm month-1 for Noah-MP and 0.016 cm month-1 for CLSM with p <0.01 for all three trends).  Larger discrepancies are also observed in the Congo and Chad basins (Figs.2b,l), which may be linked to deforestation and are discussed in more detail in section 3.2.*

*Overall, the reanalysis TWS captures interannual variability observed by GRACE/FO but tends to underestimate strong anomalies such as the 2019 elevated TWS in the Nile basin and the 2016 and 2019 low TWS in the Zambezi basin, reflecting deficiencies in land surface model physics and uncertainties in the reanalysis forcing data.  Because droughts and floods are relative to the climatological mean, evaluations in the following sections focus on non-seasonal TWS forecasts, i.e., TWS anomalies relative to monthly means".*

Table 2. Amplitudes of seasonal TWS and temporal standard deviation of non-seasonal TWS for the six largest river basins in Africa (see Supplementary Fig.S1a for basin delineations).

| | Amplitude (cm) | | | Temporal Std (cm) | | |
|---|---|---|---|---|---|---|
| | Noah-MP | CLSM | GRACE/FO | Noah-MP | CLSM | GRACE/FO |
| Congo | 4.51 | 6.52 | 7.21 | 1.53 | 5.56 | 3.23 |
| Niger | 14.83 | 21.64 | 20.81 | 1.54 | 2.29 | 3.85 |
| Nile | 5.05 | 7.41 | 7.43 | 2.13 | 2.98 | 4.71 |
| Orange | 2.66 | 3.58 | 2.44 | 2.30 | 1.68 | 2.16 |
| Zambezi | 18.27 | 24.76 | 28.87 | 2.55 | 3.54 | 5.37 |
| Chad | 11.74 | 16.97 | 16.62 | 1.31 | 2.70 | 2.49 |

- L. 224: Why are here 2 numbers. I an skeptical that trends of few micrometers / month are significant.

  One number is for CLSM, and another is for GRACE.  This sentence has been revised to: "*The strong correlation between CLSM TWS and GRACE/FO data is also contributed by the presence of statistically significant secular trends in the two time series (p <0.01 based on the Mann-Kendall test; Yue et al., 2002), 0.014 cm month$^{-1}$ for CLSM and 0.007 cm month$^{-1}$ for GRACE/FO*".

- L 226: But I think you canceled regions with large groundwater withdrawals out. And their impact cannot be that big averaged over the entire study region.

  Good point.  The mask by Rodell et al. (2008) includes only those regions exhibiting sustained TWS declines due to groundwater withdrawals for irrigation.  Regions with intermittent groundwater use are not excluded.  Regardless, this sentence has been removed.

- L. 235: You also clearly see Lake Volta here, which shows that you cannot neglect surface water in general.

  Thanks for bringing up Lake Volta.  We have added it to the list of lakes and revised this paragraph extensively for clarity (see our response to your L.154 comment).

- L. 240: you do not show discrepancies among NMME models in S1,2 directly, maybe compute the ensemble spread?

  What we meant was discrepancies in interannual variability across NMME models.  We have revised this section extensively and this statement has been removed.

  We present interannual variability of NMME precipitation because its spatial pattern is similar to that of correlation.  In addition, RMSEs and correlations for individual NMME models exhibit clear dependency on interannual variability of precipitation hindcasts.

TWS responds to cumulated precipitation, and ensemble spreads at a specific lead time may not explain some of statistical results.

- L. 242: what about Lake Volta? It is clearly visible (however, before in the manuscript you say that surface water can be neglected).

We added Lake Volta. Thanks for pointing this out.

- L. 263: on average,··· → this is a repition.

This has been revised to "On spatial average".

- L. 283 – 286: this seems to be a central insight of your experiments. Could you highlight it a bit more?

Discussion of these results has been moved to section 3.3 and expanded considerably:

*"Since TWS hindcasts differ from the reanalysis only in the meteorological forcing data used, evaluating TWS forecasts against the reanalysis helps isolate impacts of NMME hindcasts from those of model physics. RMSEs relative to the reanalysis exhibit similar spatial patterns between the two models (Supplementary Fig.S5), with larger errors in wetter central Africa and smaller errors in drier northern and southern regions, highlighting uncertainty in NMME precipitation hindcasts, which is proportional to mean annual precipitation, as a main driver of TWS forecast errors when uncertainty in model physics is masked. As expected, these RMSEs are substantially lower than using GRACE/FO as reference, and they increase steadily with increasing lead time, reflecting growing discrepancies between NMME precipitation hindcasts and CHIRPS precipitation estimates.*

*Correlations evaluated relative to the reanalysis reveal contrasting patterns between the two models (Fig.S6). For Noah-MP, correlations relative to the reanalysis exhibit spatial patterns similar to those obtained using GRACE/FO as reference, with higher correlations in the drier northern and southern regions and lower correlations in central Africa. In contrast, when evaluated against the reanalysis, CLSM yields stronger correlations in central Africa through the 1–6-month lead times, in sharp contrast to the low skill inferred when GRACE/FO is used as reference. This region of strong correlations coincides with that of strong long-term TWS trends in CLSM (Supplementary Fig.S2) which, as discussed in section 3.4, may induce strong persistence in simulated TWS and hence strong correlation. For both models, domain-averaged correlations relative to the reanalysis are more than twice those relative to GRACE/FO in most cases, underscoring substantial uncertainty in model physics that limit TWS forecast skill and the need to use independent data for evaluation".*

- L. 289-291: I do not understand the relationship between decreased RMSE and overestimation of TWS interannual variability. Could you please clarify this?

*We revised to say, "In contrast, RMSEs of CLSM forecasts decrease with lead time, except for forecasts driven by GEOSv2. As shown in previous studies (e.g., Li et al., 2019b), CLSM has the tendency to overestimate TWS dynamic ranges. As interannual variability of NMME precipitation forecasts generally decreases with increasing lead time (Supplementary Fig.S3), elevated dynamic ranges are suppressed, leading to reduced RMSEs at longer lead times. In contrast, GEOSv2 precipitation exhibits increasing interannual variability with increasing lead time (Supplementary Fig.S3), eliciting amplified TWS changes and thus larger RMSEs at longer lead times".*

- L. 294 – 295: model physics have stronger influences than meteorological forecasts: didn't you show that this is different for NOAH and CLSM?

  Because the two models are driven by the same forcing dataset, the different behaviors in RMSEs, one increasing with lead time and another decreasing with lead time, suggest stronger role of model physics than meteorological forecasts. Regardless, this sentence has been removed with the revision.

- L.305: So would it be better to use the forcings that lead to the best results for forecasts and not the ensemble mean?

  The model performance varies across the continent, making it unlikely that a single NMME model performs best in all regions. Also, FLDAS-Forecast is used for soil moisture and streamflow forecasting which may respond differently to NMME precipitation. Nevertheless, this issue warrants further investigation and can be addressed in future studies.

- L 326: Could you define persistence somewhere in one sentence?

  We define persistence in the first sentence of the paragraph as "*Persistence refers to the tendency of a process retaining its past states (wet or dry conditions) and has been known to help enhance hydrological prediction skill. To examine persistence, we computed the autocorrelation of TWS time series from the two re-analyses and GRACE/FO data at three lags*".

- L 358: Could you have an introductory sentence to remember the reader which kind of percentiles you are referring to?

  Yes. We added this sentence at the beginning: "*To demonstrate the value of TWS forecasts, we examine TWS percentile maps derived using equation (1) for hindcasts initialized in December 2015 and GRACE/FO data (Fig.9)*".

- L 358ff: would it make sense to involve percentiles from GRACE into the extreme event discussion?

  Yes. GRACE based percentiles have been added. Please see our response to Review#1.

- Fig. 8: Why do you show and discuss percentiles only for CLSM and not for NOAH?

  Noah-MP forecast percentiles have been added. Please see our response to Review#1.

**4 Summary and discussion**

In this section many aspects from the results section are repeated. It would be great if you could add in each paragraph some more interpretation or insights what this means for future research or applications.

This section has been revised extensively to provide more insights and synthesis in several key areas, including:

1) relevance of interannual variability to S2S forecasts
2) performance of CLSM vs that of Noah-MP
3) uncertainties in model physics and the importance of improving groundwater simulation in land surface models
4) relationship between interannual variability and uncertainties in NMME precipitation hindcasts
5) limitations of persistence for predictability, and importance of using independent datasets to evaluate forecasts

We also explicitly call on future research to develop surface water datasets harmonized with GRACE data to assist separation of surface water from the record and quantify associated uncertainties.

**Minor comments:**

**Abstract:**

- Abbreviation FEWS NET not defined.

 Defined.

**Data and evaluation metrics:**

- Table 1: abbreviations are not defined

 Definitions are provided in the table caption.

**Results**

- L 225: show a statistical··· → add "a"

Done

---

## Author Comment (AC2)

**Review #1**

**Summary**: the authors offer an evaluation of FEWSNET S2S terrestrial water storage forecasts for Africa. The manuscript focuses on differences between the two land surface models included in the FEWSNET forecast ensemble--CLSM and Noah-MP--and offers commentary on the performance of each. Overall, they conclude that CLSM offers advantages when simulating and forecasting TWS. Results also show how various NMME meteorological S2S forecasts compare, but these results are not emphasized in the discussion. The primary source of evaluation data in the main text is GRACE, while information on precipitation forecasts is contained in supplementary material and is addressed only briefly in the text.

I find the results presented in the manuscript to be interesting, and the explanation of these results is generally quite clear and useful. I did find myself a bit confused at times, when the authors bounced between comparing hindcasts to reanalysis and comparing hindcasts to GRACE observations, and when some of the explanation of geographic patterns seemed to me to be speculative. But these are minor points, and I have only a few questions that I would like to see addressed before the paper is published in final form.

Thanks for your supportive comments.  We agree that our initial submission overlooked some important findings related to precipitation forecasts and we have since revised the relevant paragraph in the Summary section to highlight those findings: "*Consistent with the above discussion, TWS forecasts are highly sensitive to inter-annual variability of precipitation forecasts, which differs substantially across NMME models.  TWS forecasts driven by precipitation forecasts with larger interannual variability (e.g., GEOSv2) showed lower correlation and higher RMSEs with respect to GRACE/FO observations, whereas those driven by precipitation forecasts with lower interannual variability (e.g., GFDL and CSM5) yielded more accurate TWS forecasts.  Performance of TWS forecasts also responds to changing interannual variability of NMME precipitation forecasts with lead time.  In most cases, precipitation forecasts exhibit decreasing interannual variability with increasing lead time, likely reflecting reduced forecast skill at long leads when prediction reverts toward climatology (Zhang et al., 2021). This decrease in variability leads to contrasting model behaviors, with CLSM TWS forecasts showing reduced RMSEs at longer lead times, whereas Noah-MP forecasts exhibit increasing RMSEs*".

In addition, we created a new section, section 3.3, to present statistical results relative to the reanalysis to avoid confusion between the two sets of evaluation statistics.  Both the

Results and Summary and discussions sections have been extensively revised to improve accuracy and reduce redundancy.

**Specific comments**:

Line 204: isn't the 1m CLSM "soil depth" a choice that was made by the authors? This implementation of the model might output 1m soil moisture, but the model also has an implicit soil water profile that could be used to extract an estimate of total soil moisture integrated to any depth. Similarly (and maybe more easily) the authors could have used 1m soil moisture from Noah-MP rather than the full 2m column. Why not compare 1m CLSM to 1m Noah-MP, or 2m CLSM to 2m Noah-MP?

The 1 m CLSM root zone depth is prescribed by model developers, not a choice made in this study. The model only has three subsurface states, a 2 cm surface layer, a 1 m root zone and the total profile. Indeed, the profile soil water includes deeper soil moisture. But because it does not explicitly simulate groundwater, which is also included in the total profile soil water, there is no other way to extract deeper soil moisture.

Comparing 1 m CLSM with the 1 m root zone soil moisture from Noah-MP would yield more comparable dynamics. However, as the purpose of this analysis is to assess the relative contributions of soil moisture and groundwater to TWS dynamics, we want to present soil moisture in the entire unsaturated zone.

We clarified this issue in section 2.3 as follows: "*Although CLSM does not explicitly model groundwater, groundwater variation is included in the total profile soil moisture; thus, CLSM groundwater storage is obtained by subtracting water storage in the root zone from that of the total soil profile, following previous studies (e.g., Li et al., 2019b). Compared to Noah-MP, CLSM groundwater contains soil moisture from the 1-m depth to the implicit water table. Despite this diagnostic approximation, CLSM groundwater has been shown to compare well with in situ groundwater in different climates (Xia et al., 2017; Li et al., 2019b)*".

Lines 234-249: In Figure 2, the reanalysis errors look almost identical to the forecast errors for both Noah-MP and CLSM. Yet the authors invoke NMME uncertainties when explaining some aspects of model errors. Given that the patterns and magnitude of error appear to be very similar in reanalysis and in forecasts at all lead times, aren't these errors more about model bias than about forecasts? Even the explanations that invoke interannual climate variability seem like they'd need more evidence in their support, since we'd want to know that errors in interannual meteorological variability are seen in a similar way in both CHIRPS (or MERRA-2) and in the NMME models.

Thanks for this comment. The similarity in RMSEs between the forecasts and re-analysis was discussed in Line 250, but it may have been overlooked. We agree that invoking interannual variability in NMME precipitation here is inappropriate and that uncertainties in model physics contribute substantially to those large RMSEs. However, given that there is also similar RMSE patterns between the two land surface models such as the large RMSEs in southern Zambia and Angola, precipitation errors also likely played a role in those spatial patterns of RMSEs.

We revised the opening paragraph of section 3.2 as: "*RMSEs of the ensemble mean TWS hindcasts of all NMME models, with respect to GRACE/FO data, exhibit distinct spatial patterns (Fig.3). Large RMSEs are observed in the interior western Sahel, a large region across Lake Victoria, Lake Tanganyika, and Lake Volta as well as southern Zambia and Angola, for both models. As the models do not simulate surface water which is detected by GRACE/FO satellites, unresolved surface water dynamics and water management activities may have contributed to errors in lake areas. In addition, uncertainties in precipitation forcing data, for both reanalysis and hindcasts, especially under a changing climate, may further amplify errors in simulated TWS. As discussed earlier, the East African Rift, which includes Lake Victoria, has seen increased precipitation variability (Boergens et al., 2024); similarly, Southern Africa including southern Angola has been experiencing erratic precipitation patterns and more severe meteorological droughts in recent years (Trisos et al., 2022; Correia et al., 2025). However, considering that the reanalysis exhibits similar spatial patterns and magnitudes of RMSEs as the hindcasts (Figs.3a,e), deficiencies in model physics are likely the dominant contributor to RMSEs in TWS hindcasts*".

Line 285: If these results compare model forecasts to their own reanalysis, can we really say that degradation of Noah-MP forecasts is due to an "inability" to simulate long-term TWS variability? Couldn't we just as easily say that the persistence of CLSM forecasts is due to that model's "inability" to simulate rapid runoff and drainage? Without an independent evaluation dataset (for this specific result) it's not possible to know which model's behavior is better. That said, the subsequent results that *do* offer comparison with GRACE make a more convincing case. I would recommend that the authors avoid making statements about the quality of model performance when using the retrospective simulations as the truth. (In fact, they might consider moving these statements out of this section, as I admit that I was confused on my first reading about which statements had an observational basis and which were about simulation comparisons.)

Your points are well taken.  We moved the statistical results relative to reanalysis to a new section, section 3.3.  We also eliminated words like "inability" when discussing results using the reanalysis as reference.  We revised this section extensively to simply describe the results without speculation.  We also added overestimation of surface runoff as an additional factor impacting groundwater dynamics in the Summary and discussions section.

Section 3.4: Why aren't any GRACE comparisons offered in this section? It seems odd to show the forecast without any evaluation.

Noah-MP and GRACE based percentile maps have been added in Fig.9 (previously Fig.8, see below image) and corresponding discussions have been included in this section.

[Figure]

Fig.9 TWS percentile maps derived from Noah-MP and CLSM mean TWS forecasts (top two rows) of all NMME models, initialized in December 2015, and corresponding maps for GRACE/FO data (bottom row).

---

## Author Comment (AC3)

**Review#3**

The article evaluates S2S TWS forecasts produced from FLDAS over Africa using gravity observations. I think the article reads well and I think it stretches the surface of a relatively unexplored area. That is, it highlights the importance of improving model physics of groundwater as well as it relevance for S2S forecasts. I want to also say that the authors motivate the GRACE community to reduce latency on their products, as GRACE-DA could be beneficial to improve the forecast. I'd encourage the authors to add some of these cavetas in the conclusions section. Beside this "major" comment, I, here, list only minor suggestions for the authors.

Thank you for your supportive comments. We revised the final section extensively to highlight the importance of improving groundwater simulation for enhancing TWS forecasts. We also emphasize the need for reducing GRACE data latency to benefit TWS forecast in the last section.

2.2. > it is unclear to me what NMME models are, at around line 111, please add a brief broad description of why they are needed here. Also unclear what and why the downscaling is needed. Table 1> what variables of these models are used?

Background information on NMME models and why they are needed have been provided in section 2.2 as the following: "*To generate TWS hindcasts, atmospheric forcing fields must be obtained from hindcast products to properly represent forecast uncertainty, rather than from reanalysis which cannot predict future weather events. Unlike reanalysis, meteorological hindcasts are produced by climate models without constraints of observations and therefore, are subject to larger uncertainties. FLDAS-Forecast employs a suite of NMME models developed by multiple institutions to provide S2S precipitation (and temperature which is not currently used by FLDAS-Forecast) forecasts (Table 1). The ensemble approach not only enables uncertainty quantification but also generally yields higher predictive skill than any single model (Wood et al., 2002; Kirtman et al., 2014)*".

Downscaling is needed because of the coarse spatial resolutions of the hindcasts, "*NMME precipitation hindcasts are provided as monthly data on a 1° global grid, while non-precipitation GEOS hindcasts are provided at 0.5° in latitude by 0.625° spatial resolution. All meteorological hindcasts are bias-corrected and spatially downscaled to the 0.25° resolution using CHIRPS and MERRA-2 data, respectively, and further temporally disaggregated using LIS built-in functions (Arsenault et al., 2020; Hazra et al., 2023)*".

Only the precipitation field from NMME is used in the FLDAS forecast system. We updated the table caption to clarify this.

Line 138: Typo CLMS

Corrected.

Line 186: are the percentiles computed using seasonal mean? Please clarify in manuscript

Yes. This has been clarified in section 2.5 where an equation for computing percentiles has been added.

Fig 4. Why is the correlation so small already at 1-month lag time? How significant are these statistics?

Thanks for the question. As shown in the spatial correlation maps, there are regions of high correlations (see below). However, low and even negative correlations in certain areas (due to the opposite trends in simulated and GRACE observed TWS) suppressed domain-average correlations. We have updated the correlation and other maps using grey to mask out groundwater depletion regions and a triangle is used for the low end of the label bar (see below) to highlight that any values lower than -0.1 are shown in white.

The manuscript has also been updated to say, *"In CAR and South Sudan, TWS hindcasts from both models show near and below zero correlations with GRACE/FO data, due to the opposite trends between reanalysis TWS and GRACE/FO data in that region (Supplementary Fig.S2)"*.

[Figure]

*Fig.4 Correlation between non-seasonal reanalysis TWS and ensemble mean TWS forecasts of all NMME models at three lead times, and GRACE TWS observations for Noah-MP (top row) and CLSM (bottom row). Domain average correlations are shown in inset text.*

We didn't do significance test as the purpose was to compare the correlation between the two models.  Significance test also requires temporal independence of the data, which is not the case for TWS.

Line 308 – 324: it is unclear in my option what this analysis is really telling us. What is ROC and why is it computed only on the lower tercile (drier forecasts?)? Please add some general background on the metric and its interpretation.

Before discussing ROC results, we added the reason why we use ROC scores, "*While RMSEs and correlation quantify the magnitude of discrepancies and the temporal consistency between two time series, they do not directly assess the ability to accurately forecast wetter and drier conditions.  Therefore, we use ROC scores to evaluate the performance of Noah-MP and CLSM in predicting terciles, corresponding to below-normal, near-normal and above-normal conditions*".

In addition, we describe ROC in the data section (section 2.6), "*Additionally, skill in forecasting terciles is assessed using the relative operating characteristic (ROC) score, a commonly used evaluation metric measuring the ratio of hit rates to false alarm rates (Met Office).  A ROC score of 1 indicates a perfect forecast. ROC scores below 0.5 suggest no*

*skill, while scores above 0.6 indicate predictive skill (Met Office). High ROC scores and strong correlation are commonly interpreted as indication of skillful forecasts (e.g., Yuan and Zhu, 2018)".*

The spatial map for upper terciles is provided in the supplementary file (see Line 316 of the manuscript) as it is very similar to that of lower terciles.

Fig 6 > wouldn't be useful to also show difference maps of the first two rows wrt GRACE (bottom)?

Thanks for the suggestion. We now show differences in persistence (see below image) and added the following text to the manuscript," *Differences between simulated and GRACE/FO TWS persistence exhibit spatial patterns similar to those of mean annual precipitation, reflecting the strong influence of precipitation and associated uncertainty on persistence (Fig.7, bottom two rows). However, the two models often exhibit contrasting performances. Compared to GRACE/FO TWS, Noah-MP underestimates persistence in central Africa and overestimates it elsewhere. In contrast, CLSM overestimates persistence in central Africa while underestimating it in other regions, a discrepancy that is more pronounced at the 2- and 4-month lags. In wetter central Africa, strong interannual variability in CLSM TWS helps retain past wetness conditions and contributes to its enhanced persistence. On the other hand, the underestimation of persistence by CLSM in drier regions may be linked to the model's tendency to overestimate ET, which acts as continuous disruption to soil moisture states, thus leading to low persistence. For both models, discrepancies between reanalysis and GRACE/FO increase with increasing lags (Fig.7, bottom two rows), reflecting cumulated differences in their abilities to retain past states".*

[Figure]

*Fig.7 Autocorrelation of Noah-MP and CLSM reanalysis TWS (top two rows), and GRACE/FO data (third row) at three lags. The fourth and fifth rows show differences in autocorrelations between the reanalysis and GRACE/FO data. Upper right text in the top three rows shows average autocorrelation and fraction of area with autocorrelation>0.37 (in parentheses).*

Fig 8 . Can the authors make it clear that top left figure is the IC and everything else is forecasts? At first I thought top raw was initialization, while the bottom was the forecasts.

The layout of this figure has been improved, and percentile maps of Noah-MP forecasts and GRACE data have been added. IC and forecast are now clearly labeled in the top two rows.  Please see our response to Review#1 on update of this figure.